# Fermented Foods in the Management of Obesity: Mechanisms of Action and Future Challenges

**DOI:** 10.3390/ijms24032665

**Published:** 2023-01-31

**Authors:** Mahsa Jalili, Maryam Nazari, Faidon Magkos

**Affiliations:** 1Department of Nutrition, Exercise, and Sports, University of Copenhagen, 1165 Copenhagen, Denmark; 2Food Safety Research Center (Salt), Semnan University of Medical Sciences, Semnan JF62+4W5, Iran

**Keywords:** obesity, fermented foods, immunity, metabolism, gastrointestinal microbiome

## Abstract

Fermented foods are part of the staple diet in many different countries and populations and contain various probiotic microorganisms and non-digestible prebiotics. Fermentation is the process of breaking down sugars by bacteria and yeast species; it not only enhances food preservation but can also increase the number of beneficial gut bacteria. Regular consumption of fermented foods has been associated with a variety of health benefits (although some health risks also exist), including improved digestion, enhanced immunity, and greater weight loss, suggesting that fermented foods have the potential to help in the design of effective nutritional therapeutic approaches for obesity. In this article, we provide a comprehensive overview of the health effects of fermented foods and the corresponding mechanisms of action in obesity and obesity-related metabolic abnormalities.

## 1. Introduction

Obesity is a highly prevalent disease globally and increases the risk for other chronic conditions like type 2 diabetes mellitus, coronary heart disease, osteoarthritis, sleep apnea, and some types of cancer [1]. Anti-obesity drugs are often used to prevent and treat obesity and its comorbidities; however, various side effects and inflated healthcare costs have resulted in seeking safe and effective natural product alternatives in obesity management [2]. Obesity is a multi-factorial health challenge that is accompanied by immune dysfunction of white adipose tissue and intestinal microbiome dysbiosis; recently, the administration of bacteria or their metabolites from fermented foods has been found to beneficially affect adipose tissue function, inflammation, and the gut microbiome [3,4,5,6,7].

Fermentation has been used for centuries to preserve foods and their texture through deactivating spoilage microorganisms [8,9]. Fermented foods contain enduring microorganisms that predominantly include lactic acid bacteria (LAB) and their major metabolites, i.e., lactate [9,10]. These foods are thought to have health-promoting effects due to a mixture of several beneficial microorganisms—not only LAB species but also acetobacter and Propionibacterium; and bioactive macromolecules—such as exopolysaccharides and bactericides [11,12,13,14]. However, the mechanisms of action are not entirely clear [10,15].

This paper aims to provide a comprehensive overview of available evidence from recent clinical and experimental investigations on fermented foods and obesity, focusing on the challenges faced, knowledge gaps, and future perspectives, and the potential molecular mechanisms by which fermented foods affect the gut microbiome and immune responses.

## 2. Fermented Foods, Gut Microbiota, and Metabolic Regulation

Fermented foods function as a relatively optimal drug delivery system as they contain a mixture of safe bioactive compounds in sufficient doses to transfer to the target site, thereby facilitating efficacy and long-term compliance [11,16,17]. The food matrix of fermented foods can exert medicinal properties by a large number of viable microorganisms and their transformed metabolites to shield the gastrointestinal tract against pathogens, excess gastric acid, and bile salts [18,19]. Probiotic microorganisms in fermented foods are capable of producing short-chain fatty acids (SCFAs) via fermentation of the prebiotic non-digestible carbohydrates, which can then be taken up as a source of energy by bacteria in the colon, inhibit the overgrowth of intestinal pathogens, and regulate various metabolic pathways (e.g., cholesterol synthesis), including the secretion of appetite hormones [9,20,21,22,23].

Fermented foods also contain live microorganisms or beneficial bioactive compounds that support the symbiosis between the host and the microbiome, resulting in a healthier gut environment [10,21]. SCFAs like acetate, propionate, and butyrate and other primary metabolites of live probiotics from fermented foods can stimulate the growth of beneficial microbial phyla in the gut, for example, by lowering intestinal luminal pH and improving conditions for intestinal commensal microflora like Bacteroides and Prevotella species [22,23].

The association between the gut microbiome and the risk of obesity has been known for some time [24,25,26]. Dysbiosis refers to an imbalance of intestinal microflora that is associated with obesity [20,27]. Several experimental and clinical studies have demonstrated that a lower ratio of Bacteroidetes to Firmicutes may be involved in the pathogenesis of obesity; however, due to large inter-individual variation this ratio cannot be considered as a biomarker for increased obesity risk, and further research is required to identify a key bacterial community based on individual characteristics and traits that increases susceptibility to weight gain and body fat accumulation [27]. Although the abundance and composition of healthy gut microflora are different depending on geographical region [28], decreased diversity of the microbial populations is often seen in obese subjects [29]. It remains to be confirmed if manipulation of the microbial community in the gut can result in a long-lasting effect on appetite regulation and body weight homeostasis. 

Intestinal permeability is another factor that may mediate some of the associations between the gut microbiome and obesity. The microbiome interacts with inflammatory signal transduction pathways via the gram-negative bacteria membrane lipopolysaccharide (LPS), which binds to CD-14 and toll-like receptor-4 (TLR4) on enterocytes, resulting in the translocation of bacteria through the intestinal barrier and an inflammatory response [27,30,31]. Relevant studies indicate that gut microbiome dysbiosis can increase intestinal permeability that can then lead to immune cell infiltration from the intestinal epithelium to the white adipose tissue and initiate low-grade systemic inflammation [32]. Increased abundance of the protein zonulin in feces is a marker of abnormally increased gut permeability and disturbed gut barrier function. Recent evidence demonstrates that fermented dairy products, such as kefir, can reduce intestinal permeability and gut tight junction dysfunction [33,34], although this does not necessarily result in beneficial changes in serum pro-inflammatory markers [33]. 

The bioactive compounds in fermented foods can also exert beneficial effects on adipose tissue function via upregulation of the peroxisome proliferator activator receptor γ 2 (PPARγ2) [24]. Excess fat accumulation in white adipose tissue and reduced lipid turnover lead to increased macrophage infiltration and pro-inflammatory cytokine overproduction that predispose to metabolic dysregulation. PPARγ2 suppresses resistin expression in white adipocytes and improves insulin sensitivity [25]. Activation of PPARγ2 also reduces plasma free fatty acid concentrations (partly because of enhanced insulin-mediated suppression of lipolysis) and improves the plasma lipid profile. Furthermore, it suppresses pro-inflammatory mediators and increases macrophage function in visceral adipose tissue [35].

Understanding the effects of fermented foods on the composition and function of the microbiome can contribute to better nutritional therapies for long-term body weight homeostasis. Since many individuals with obesity can lose weight but cannot maintain it for prolonged periods of time, finding safe and long-term dietary treatments has been of great interest [36]. Fermented foods produce SCFAs in the gastrointestinal tract that can regulate the intestinal microbiome, inhibit inflammatory pathways, and reduce appetite hormones [10,20]. Furthermore, fermented foods may contain live microorganisms or beneficial bioactive compounds that support the symbiosis between the host and the microbiome, resulting in a healthy gut environment [10,21].

SCFAs, including acetate, propionate, and butyrate, are the products of fermentation of dietary non-digestible carbohydrates by gut bacteria. They can inhibit lipid synthesis enzymes, reduce pathogen microorganisms, and supply energy for the intestinal epithelium [37]. SCFAs and other primary metabolites of live probiotics from fermented foods can stimulate the growth of beneficial microbial phyla in the gut; for example, they lower intestinal luminal pH and improve conditions for intestinal commensal microflora like Bacteroides [23] and Prevotella [22] species.

## 3. Mechanisms of Action of Fermented Foods in Obesity

There are various types of traditional fermented foods including dairy products, fruit juice, and foods of mixed composition. Although the molecular mechanisms for the observed clinical outcomes linked to consumption of distinct types of fermented food products are too broad to classify in just a few general categories, the major modes of action are summarized in the following sections and are illustrated in Table 1. To include data on the efficacy of specific fermented foods on body weight, fat mass and food intake, studies were selected according to the primary outcome considering the biological pathways relevant to metabolic regulation, immune activation, and intestinal microbiome modulation. Several molecular biomarkers have been studied in the selected clinical trials, experimental animal studies and in-vitro experiments. In most of the included studies, multiple mechanisms of action were evaluated in response to interventions with fermented food products.

### 3.1. Inhibition of Lipid Synthesis in the Liver and Other Metabolic Organs

Current evidence suggests some lactic acid bacteria, such as Lactobacillus plantarum, have anti-obesity effects; however, their mode of action is not adequately identified. A mixture of L. plantarum species can reduce fat synthesis and storage, the gene expression of adenosine monophosphate-activated protein kinase-α (AMPK-α), fatty acid synthetase (FAS), acetyl CoA carboxylase (ACC), and peroxisome proliferator-activated receptor-γ (PPAR-γ). At the same time, serum concentrations of total and LDL-cholesterol and triglycerides decrease and HDL-cholesterol increase [45]. Among over 400 strains of LABs, the Lactobacillus plantarum strain Ln4 and fermented foods containing this LAB inhibit lipid storage and adipocyte differentiation, body weight, lipid accumulation, and insulin resistance through suppression of adipokine proteins such as ANGPT-L3 (angiopoietin-like), C-reactive protein (CRP), leptin, lipocalin-2, monocyte chemoattractant protein-1 (MCP-1), and insulin-like growth factor binding proteins (IGFBPs) in white adipose tissue. They also stimulate glucose uptake by upregulation of the gene expression of hepatic lipid metabolism factors including lipoprotein lipase (LPL), insulin receptor substrate 2 (IRS2), protein kinase Bβ (Akt2), and AMPK in preadipocytes. Ln4 is a common probiotic in kimchi, a Korean fermented food, and was separated from kimchi culture that has anti-diabetic and anti-obesity effects in experimental studies [50]. Other types of traditional fermented foods also exhibit lipid-lowering effects. Another common fermented food in Korea is kochujang. It is fermented red pepper, soybean, and rice. Fermented kochujang can upregulate acyl-CoA synthetase (ACS), carnitine palmitoyl transferase-1 (CPT-1), and uncoupling protein-1 (UCP-1) and downregulate ACC gene expression; these changes in gene expression and subsequent protein function can result in weight reduction [62]. 

Pumpkin (Cucurbita moschata) seeds have been used for centuries in the Native American medicinal food culture for the reduction of blood glucose, cholesterol, hypertension, and the treatment of intestinal infections and disorders [81]. Fermentation is an effective treatment to reduce anti-nutritional factors of fresh pumpkins. The fermented pumpkin with LAB reduces body weight, body fat content, plasma lipid profile, and liver enzymes by suppressing gene expression of lipogenic factors including PPAR-γ, CCAAT-enhancer-binding proteins (C/EBPα, C/EBPβ, C/EBPγ), and sterol regulatory element-binding transcription factor 1 (SREBP1C) [52]. Not only non-pathogenic LAB but also fungi exhibit beneficial effects on metabolism. For example, koji, a Japanese cultivated cooked rice with the non-pathogen fungus Aspergillus oryzae contains high levels of a growth-boosting metabolite named glucosylceramide, which enhances some beneficial gut bacteria like Blautia coccoides [82]. Koji glycosylceramide (KGC) increases gene expression of CYP7A1 and ABCG8, which are important in cholesterol catabolism and its excretion in the form of bile acids [83]. Fermentation with aspergillus strains produces metabolites like free sugars, amino acids, proteins, fibers, and polyphenolic compounds [84]. Okara is a Japanese soy food product that has been fermented by probiotic aspergillus strains from the koji culture [85]. Short-term consumption of okara can reduce body weight, fat content, improve blood and liver lipid profile, and lipid metabolism; however these changes occur without a significant effect on SCFAs in the intestine [39].

Concurrent intake of antioxidants and fermentation metabolites can potentially boost metabolic effects. For instance, fermented gallate-containing green tea inhibits adipocyte differentiation without induction of apoptosis and reduces body weight without any change in calorie intake and hepatic cytotoxicity, suggesting that these effects are not dependent on appetite regulation and liver damage pathways [49]. Regardless of whether the fermented food contains live probiotic microorganisms or not, beneficial effects have been demonstrated in several experimental and in-vitro studies, thus implicating several metabolites from fermented food in the regulation of lipid metabolism pathways in the liver and adipose tissue that eventually lead to weight reduction.

### 3.2. Reduction of Appetite Hormones

Appetite control is a key factor in obesity management. Food intake and total energy expenditure are important for long-term body weight homeostasis and can be regulated by multiple physiological pathways [86]. Peripheral regulation of appetite consists of adipose-related factors including insulin, leptin, and adiponectin, as well as a variety of intestinal peptide hormones secreted in response to food ingestion [86]. Besides total calorie intake, nutrient selection can influence the regulation of hunger and satiety. Various nutrients in the food matrix produce a complex set of metabolites in the gastrointestinal tract that can affect liver, intestine, and adipose tissue metabolism and can subsequently have a profound impact on appetite-related neurotransmitters in the hypothalamus [87]. 

Fermented dairy products slow gastric emptying, so they can influence postprandial responses like those of blood glucose, insulin, and lipids. Sangaard et al. reported a lower gastric emptying effect of fermented milk A38 due to its higher viscosity. A38 intake resulted in a greater increase but also a faster decrease in serum triglyceride concentration in all lipoprotein fractions. The same trajectory pattern (greater peak but faster decrease) was observed for cholecystokinin (CCK) and peptide YY (PYY), gastric inhibitory polypeptide (GIP), and glucagon-like peptide-1 (GLP-1). Increased GIP levels after A38 consumption can be a result of GIP response to dietary intake of fat in conjunction with slower gastric emptying. The increased postprandial CCK levels can be related to the response to the high content of whey protein. Although satiety hormones were significantly elevated, the subjective satiety sensation (self-reported on a visual analogue scale) did not change [64]. Likewise, a study on the satiety effects of fermented soy meal (tempeh) on appetite hormones and subjective appetite scores reported significant effects on ghrelin, insulin, and arginine, but the satiety, fullness, and hunger were not different between fermented and unfermented meals [41]. By contrast, a study with propionate fermented milk drink reported a significant effect on fullness, hunger, and calorie intake; however, this was a short-term effect of up to 50 min after eating [88]. 

Another interesting study evaluated the role of fermented food as a compound delivery system. The researchers investigated whether emulsified lipid Fabuless (Olibra) can reduce dietary energy intake and improve weight management. The findings revealed a non-significant effect of Olibra, separately or with solid food, on postprandial satiety sensation. On the other hand, Olibra plus yogurt was effective in reducing appetite. The anorectic effect of Olibra was enhanced with yogurt consumption by a longer transit time from the mouth to the large intestine [89]. 

SCFAs are the commonest metabolites of fermented foods, and they have an appetite-suppressing effect [90,91]. The mechanism of action of SCFAs like acetate in appetite control is not only through activation of acetyl CoA carboxylase (ACC) and peripheral inhibition of energy intake by satiety neuropeptides, but also through a central modulation of the hypothalamus and GABAergic neurons to regulate appetite [61]. Johansson et al. evaluated the appetite-reducing effect of fermented whole grain bread compared to unfermented whole grain, and refined products. Both fermented and unfermented whole grain products produced significant changes in fullness, hunger, and insulin postprandial responses [92]. In that study, the content of dietary fiber was a principal factor driving satiety, implying that the metabolites produced by fermentation were not sufficient to influence satiety. Another study from the same research team reported a significant effect of fermented sourdough rye bread on appetite. Although the mode of action is not clear, it was suggested that fermentation of sourdough can break down the viscous fibers and enhance the bolus dissolution compared to yeast-fermented rye bread [7]. Nevertheless, another study by the same researchers reported the opposite result, i.e., that sourdough does not have any effect on appetite and subsequent dietary intake. Despite a significant effect of sourdough on protein aggregation, the acidity was not enough in sourdough bread samples and the content of rye in different test pieces of bread was not significantly different. The lack of an effect on appetite might be related to relatively low doses of consumed rye in all groups [93]. Due to these controversial results, it is a bit early to draw a conclusion about the effect of fermented foods on appetite hormones and weight homeostasis. Further research is needed on the role of fermented foods on appetite regulation and weight management.

### 3.3. Inhibition of Proinflammatory Cytokines

Excessive accumulation of white adipose tissue demonstrates a dysregulated pattern of secretion of adipokines associated with abnormal inflammatory cytokine profile and chronic inflammation in obesity [94,95]. The imbalance between pro-inflammatory and anti-inflammatory cytokines in adipose tissue in obesity triggers dysregulated energy homeostasis and elevated numbers of inflammatory immune cells including M1 macrophages, neutrophils, mast cells, CD8^+^ T cells, and Th1 cells [96]. A chronic inflammatory state leads to elevated lipolysis in other tissues like muscle, liver, and pancreas, eventually followed by lipotoxicity and insulin resistance [97,98].

LAB strains separated from fermented dairy products have anti-inflammatory and anti-obesity effects. A mixture of LAB strains from Mozzarella di Bufala Campana (1 × 10^9^ CFU/day) increased the number of immune-regulatory leukocytes like CD4^+^ T lymphocytes, CD4^+^ CD25^+^ Treg cells, and decreased pro-inflammatory leukocytes including CD8^+^ T lymphocytes, CD11b^+^ activated leukocytes, and F4/80^+^ macrophages during weight reduction [53].

Supplementation with probiotic yogurt containing L. delbrueckii subsp. bulgaricus Streptococcus thermophilus and probiotic Bifidobacterium animalis (1 × 10^8^ CFU/day) for 60 days elevates IgA^+^ cells in the intestine and reduces body weight. The IgA^+^ cells secrete immunoglobulin A which plays a crucial role in maintaining the efficiency of the epithelial barrier. In addition, increased levels of IL-10 could suppress inflammation in the intestinal epithelium [56].

Sichuan pickles are a traditional food in China that is fermented by lactic, acetic, and ethanol pathways, thereby leading to variable characteristics and chemical features of vegetable pickles. The most abundant metabolites in sichuan are volatile organic products. The concentration of acid, ester, aldehyde, and alkenes depends on the bacterial diversity in the sichuan fermentation process [99]. *Lactobacillus fermentum* isolated from sichuan pickle can upregulate PPAR-α and downregulate PPAR-γ, suppress adipose tissue enlargement, and improve blood lipoprotein and liver damage markers [46]. *Lactobacillus acidophilus* SJLH001 (*La*-SJLH001), another strain isolated from Chinese fermented foods, has a potential regulatory role in glucose and cholesterol metabolism in obesity and metabolic dysfunction. It reduces serum glucose and total cholesterol concentrations and downregulates pro-inflammatory genes including CD36, Reg3γ, TLR2, and PPAR-α [47]. TLR2 gene expression is associated with increased innate immune response in immune disorders and type 2 diabetes as TLR2 levels are inversely associated with glucose transport [47,100]. The downregulation of this gene can prevent metabolic dysfunction and immune complications in obesity. 

Exercise has been proposed to improve glucose tolerance and body fat-free mass; however, increases oxidative stress damage and pro-inflammatory markers in the muscle. Fermented foods in combination with exercise may exert enhanced beneficial effects on calorie intake and body fat mass as well as immune-related myokines compared to separate administration of fermented food or exercise alone [45,85]. Fermented soy alone reduces body fat and expression of TLR4, MyD88, and IL-6 in muscle cells compared to resistance exercise alone or control group. Fermented soy in addition to resistance exercise can compensate the adverse effects of training on inflammatory TLR-4, but the findings were not promising for all the pro-inflammatory factors (86).

There is also evidence that fermented foods may have a dose-dependent effect in weight control, glucose and lipid homeostasis, and innate immunity. For example, high-dose fermented mixed grains plus digestive enzymes reduce body weight, adiposity, leptin, keratinocyte chemo-attractant (KC), interleukin (IL)-1β, IL-6, tumor necrosis factor-α (TNF-α), and MCP-1 when compared to low doses [43].

The beneficial role of fermented foods is not limited to the probiotics in these products. The modified proteins in the fermented foods can regulate immune responses through inhibition of inflammatory cytokines (MCP, IL-1β, and INF-γ) in the liver as well as improved gut microbiota and gene expression patterns of adhesion molecules [101]. 

Fermentation of food byproducts may have both economical and health-related benefits. Citrus species are rich in ascorbic acid, citric acid, phenolics, and several bioactive compounds. Lemon (*Citrus limon*) peel is a byproduct that has antioxidant and anti-inflammatory characteristics [102]. Fermentation of lemon peel can increase the bioavailability and edibility of bioactive compounds, and can reduce body weight, adipose tissue, hepatic damage enzymes, and inflammatory cytokine gene expression in the liver and epididymal adipose tissue, leading to lower serum concentrations of inflammatory cytokines. HPLC analyses identified an array of different compounds in the fermented lemon peel like Vitexin, Cnidicin, and Byakangelicin, which have shown antioxidant properties that may be responsible for the observed beneficial effects [5].

Vinegar is a product of microbial fermentation that has been used as an additive in many cuisines. Long-term administration of Nipa vinegar markedly reduces weight, lipid storage, and inflammatory markers, while enhancing gut microflora and serum adipokine levels [55]. Ginseng vinegar has also demonstrated anti-obesity and anti-inflammatory effects through reduction of liver and serum lipids, TNF-α, and IL-6 [4].

Metabolites derived from fermented foods can trigger or modulate immune responses, enhance production of pro- and anti-inflammatory cytokines and chemokines, initiate microbial killing processes and generally affect signaling pathways related to oxidative stress and inflammation [103,104]. Although there is not enough evidence for every metabolite derived from fermented foods in the regulation of innate and adaptive immunity, there is a growing body of evidence suggesting an important role of these bioactive compounds in the management of obesity and obesity-related metabolic dysfunction. 

### 3.4. Improved Glucose Metabolism

Fermented foods have a stimulatory effect on glucose metabolism via upregulation of glucose transport in the intestinal epithelium and the liver, increased glucose catabolism pathways, and inhibition of oxidative stress damage [105]. The beneficial effect is not only linked to the bioactive compounds alone, but also to the health-promoting effects of microbial metabolites in the intestine that can regulate intestinal microflora and upregulate glucose transporters to attenuate insulin resistance and adipogenesis [50]. Probiotic microbes like LAB isolated from fermented foods exhibited anti-obesity and anti-diabetic roles in both in-vitro and in-vivo experiments.

*Lactobacillus plantarum* Ln4 (Ln4) is a common probiotic in Korean fermented foods and some pickles and improves oral glucose tolerance and insulin tolerance via increased cellular glucose uptake and decreased protein expression of MCP-1 and IGFBP-3, as well as decreased hepatic mRNA expression of IRS2 and AMPK.

Several plants that have been used for the treatment of diabetes may exhibit stronger effects after fermentation. Although there are few studies about the anti-obesity effects of fermented medicinal plants, the current evidence is promising. *Vaccinium angustifolium* Ait. (Canadian lowbush blueberry) has been used as a traditional treatment for diabetes in Canada. Fermentation with a natural blueberry flora bacteria called *Serratia vaccinii* enhances the biological activities of antioxidant compounds in the lowbush blueberry, including phenolic compounds, by modification of phenolic chemical structure and production of gallic acid [106]. Administration of mature fermented blueberry juice demonstrates insulin-like and glitazone-like properties on myotubes and adipocytes by activation of PPAR-γ, AMPK, and translocation of GLUT4; however, the calcium-dependent glucose transport and insulin-related glucose metabolism pathways were not affected [63]. Since overeating results in abnormally increased glucose uptake and consequently, oxidative stress damage in the adipocytes by activation of protein kinase c-δ [107], the regular intake of fermented medicinal plants and fruit juices may reverse cellular damage [44].

The bioavailability of bioactive compounds greatly influences the effectiveness of treatment with medicinal plants, as the phenolic fraction of fermented blueberry juice with *Serratia vaccinii* contains catechol, chlorogenic, gallic, and protocatechuic acids that have the greatest effect on glucose-6-phosphatase levels and glucose uptake in hepatic (H4IIE, HepG2) and skeletal muscle cells (C2C12) [108]. Considering there are various types of fermented foods that can modulate glucose uptake and metabolism, the number of studies in this field is relatively low and there is a significant knowledge gap regarding the clinical efficacy of fermented plant products in the long-term glucose homeostasis and insulin resistance.

### 3.5. Modulation of Gut Microbiome

There are several studies indicating that regular consumption of fermented foods modulates intestinal microflora, but not all studies aimed to elaborate the relevant mechanisms(s) of action in subjects with overweight or obesity. Moreover, few studies have been conducted in human subjects for an adequately long period of time to analyze the abundance and diversity of microbiota and mycobiota and their association with gut-brain axis and appetite control.

Consumption of fermented foods can increase Prevotella, Bacteroides, Lactobacillus, Leuconostoc spp., and other beneficial microbial populations in the gut. Fermented foods can also affect the gene expression of enzymes that can regulate metabolic pathways of lipid synthesis and blood pressure in the host, for example, kimchi can enhance levels of Acyl-CoA synthetase long-chain family member 1 (ACSL1) through modulation of gut microbiota that inhibits triglyceride synthesis and enhances fatty acid catabolism. In addition, increased mRNA levels of aminopeptidase N (ANPEP) are inversely associated with hypertension, angiogenesis, and inflammation; this may explain the beneficial role of regular intake of kimchi on lipid profile and blood pressure [13].

The microbial community of fermented foods likely determines their anti-obesity characteristics. Administration of a fermented food may exert significant changes in energy metabolism through modulation of the gut microbial population, including increased abundance of *Muribaculaceae*, and decreased abundance of *Akkermansiaceae*, *Coriobacteriaceae*, and *Erysipelotrichaceae* [109]. Although the modulation of gut microbiome has not been reported consistently in studies with fermented foods, the majority reveal a beneficial role in the abundance and diversity of various species in the intestine. For example, fermentation of green tea by *Bacillus subtilis* reduces *Firmicutes/Bacteroidetes* and *Bacteroides/Prevotella* ratios in obesity; in fact the *Bacteroides/Prevotella* ratio shifts towards values observed in lean subjects [110]. Inflammatory markers, lipid profile, and adiposity were changed after the intervention, and the probable mechanism underlying these observations is likely the modulation of intestinal microflora. Fermentation of herbal teas enhances some gut commensals like *Akkermansia* and *Faecalibaculum*, and reduces intestinal permeability and oxidative stress markers, in line with an improved microbiome profile [111].

Some grain-like cereals are rich in dietary fiber, protein, and minerals and have low digestibility [112]. Fermentation increases the solubility of dietary fiber and its digestibility, increases the abundance of *Proteobacteria*, *Actinobacteria*, and *Bacteroidetes,* and reduces *Firmicutes, Clostridiales*, *Lachnospiraceae*, and *Akkermansia*. The abundance of various genera of the microbiome is distinct and can be associated with the biological pathways affected by different bioactive compounds. The resistant starch content is high in some types of cereals and fermentation may enhance the beneficial effects of a functional food via reduced abundance of endotoxins and pathogens in the intestine, and changes in peripheral blood metabolites [113,114]. In some studies, although the ratio of *Firmicutes* to *Bacteroidetes* did not change significantly, the abundance of other intestinal bacterial groups such as *Akkermansia* and *Bacteroides* and SCFAs levels were augmented [43]. The administration of several types of fruit juice has beneficial effects on intestinal microbiota, but the effect of fermented juice was more distinct [115] and the correlation between the relative abundance of *Firmicutes* to *Bacteroidetes* with obesity markers was strong [116]. Not only anthropometric parameters but also several metabolic pathways such as the citrate cycle, glycerophospholipid, amino acid, and pyrimidine metabolic pathways can be improved by the modulation of the gut microbiome with fermented foods [117].

The mode of anti-obesity action of fermented foods is not only related to *Firmicutes/Bacteroidetes* and *Bacteroides/Prevotella* ratios but can also be related to other bacterial phyla like *Lachnospiraceae* and *Ruminococcaceae*, which can be affected by *Lactobacillus plantarum* HAC01 (HAC01) and *L. rhamnosus* GG (LGG) intake from fermented foods like kimchi. Despite Lachnospiraceae being a member of the *Firmicutes* family, a higher abundance ratio has been associated with the anti-obesity effects of kimchi [118]. 

Another example of altered bacterial phyla in the intestine is the *Streptococcus* genus. Intake of either fermented dairy by *Lactobacillus helveticus* or Greek yogurt increases abundance of *Streptococcus* [101]. Fermented protein consumption in both types increases gut microbial diversity and abundance, including the Streptococcus genus that is not dependent on the LDL receptor gene, although some genera, such as Akkermansia, Adlercreutzia, and Dubosiella, had a higher abundance in the wild-type genotype [119]. 

Few studies investigated the effect of fermented foods on the intestinal mycobiota as well as microbiota. For example, kefir administration to C57BL/6 mice for 12 weeks markedly augmented not only LAB, but also *Candida spp*. and *Saccharomyces spp* [71]. Although there is some evidence for health-promoting effects of probiotic yeast species, such as *saccharomyces boulardii*, the molecular mechanisms are not well studied, and more research is needed to better understand the underlying biological mechanisms. In addition, it not completely clear whether the health promoting effects of fermented foods originate from the bioactive compounds, such as dairy peptides, or the microbial content of fresh fermented food. The reason behind the beneficial roles of fermented foods through the intestinal microbiome can be related to some unknown molecular mediators that lead to anti-obesity outcomes. Figure 1 and Figure 2 summarize the main effects of fermented foods on body weight and obesity through various mechanisms in cell culture and animal studies, respectively. In Figure 1, the effects of various fermented foods on triglyceride levels and glucose uptake in the cells and the expression of inflammatory and metabolic markers have been outlined in the in-vitro cell assays. In Figure 2, the main findings of fermented foods on the anthropometric and metabolic profile, antioxidant status, and the expression of immune and metabolic markers in three different animal species have been illustrated briefly.

## 4. Challenges and Risks of Fermented Foods in the Prevention and Treatment of Obesity

Several factors, including genetic susceptibility and environmental exposure to different chemical, biological, and socio-economic factors, can influence the magnitude of response to any dietary intervention. The baseline characteristics of participants have sometimes been used to distinguish between healthy and unhealthy groups, but this may not be a reliable way of classification of inter-individual variation [10]. The relative abundance of *Bacteroidetes:Firmicutes* is a common marker to compare gut microbiota in response to lifestyle intervention in individuals with obesity, but there is no fixed value for this ratio that can be used to evaluate the effect of an intervention or differences between groups. A solution to reduce inter-individual variation can be the classification of participants according to the genetic background, dietary history, antibiotic use, etc. [120]. 

The richness and diversity of microbial phyla can change in response to ingestion of fermented foods or the combination of fermented foods with polyphenols or micronutrients. Although functional microbiology studies report significant changes after dietary interventions, it is not clear how much of this variation occurs normally and how much can be related to the effect of the administration of a specific bioactive compound or food, because the intestinal microbiome is a dynamic population, and several intrinsic and extrinsic factors play a role to maintain the balance among thousands of groups of microbial species.

Few studies have investigated the effect of fermented foods on the gut microbial eukaryote. Some concerns have been raised in this regard: the number of biological specimens, for example, feces, may be not sufficient to analyze mycobiota and targeted metagenomics assays function more efficiently in larger samples. Moreover, there is no stable and core library of fungal communities from the human gut, which makes it difficult to interpret relevant data [121]. Another point about mycobiota studies is that the variation of fungal communities in response to environmental changes or dietary interventions is low and requires a longer duration of intervention to observe significant changes. The effect of fungal metabolites on the richness and diversity of gut microbiome has not been studied well and the interaction between eukaryotic species with prokaryotic species requires further research [122].

The interventional studies with fermented foods are highly heterogeneous and there are few randomized controlled trials with a specific fermented food. This makes it difficult to pool data in a systematic manner and draw robust conclusions. Still, the heterogeneity of available clinical trials or animal experiments can be helpful to elucidate molecular mechanisms, map the pros and cons of each type of fermented food, and design better experiments for future research in this area [123]. 

The length of the experiments is also another factor that may be not sufficient to improve clinical outcomes, such as body weight and body fat mass in a 3-month or 6-month period. The primary outcomes in most studies include weight loss, fat mass, and food intake that depend on multiple factors and confounders that are difficult to control in clinical settings.

The amount of food consumed can also affect the findings. There are several doses of diverse types of fermented foods, drinks, or gavages that are not justified well and need more homogenous studies and systematic testing to pool data and make better sense of dose-response relationships.

Despite most of the experimental studies in animal models of obesity being methodologically sound, the reproducibility and generalizability of the findings have been considered to a limited extent. The precision, high throughput techniques and availability of biopsy tests and various specimens in animal models are undeniable [124]; however, most biomedical phenomena observed in homogenous animal models cannot be translated to the heterogeneous human populations or subgroups who are overweight or obese. 

Another methodological concern is the sample size calculation of the experimental and clinical studies, particularly when more than two interventions are compared among more than two groups. The power of the study might be reduced due to low sample size, and multiple test errors should be corrected by the proper statistical tests such as adjusted *p*-value post hoc tests and false positive correction tests [125]. 

It must also be noted that fermented foods are not completely safe. Fermentation of high-protein foods can produce biogenic amines that are hazardous for health. Fermented foods including dry-cured meat, fermented fish, and fermented legumes can produce N-nitroso compounds in the body, which have been linked to cancer [126]. The biogenic amines are classified into two groups: monoamines (e.g., tyramine) and diamines (e.g., histamine) [127]. Several studies have shown the detrimental effect of considerable amounts of biogenic amines in the body such as headache, increased blood pressure, stomach pain, and cancer [128]. The conservation condition of fermented foods plays a significant role in the concentration of toxic biogenic amines in the food products, for example, regular intake of canned fermented Chinese meat and soy products has been associated with gastric and esophageal cancer in low-income populations due to frequent consumption of canned fermented foods [129]. 

Another unhealthy compound in fermented foods is their salt content. The amount of salt in some traditional fermented foods, particularly Asian soy paste, and cured meat and seafood, is too high and may contribute to carcinogenesis and other adverse health outcomes [119,130]. Reduction of salt content in the traditional fermented foods can be a major challenge for food safety and palatability, as salt inhibits the growth of pathogens and production of their toxic metabolites such as mycotoxins, nitrosamines, and ethyl carbamate. In addition, salt improves the texture, aroma, and taste of fermented foods as the lower salt content is associated with higher sourness and more shapeless food texture that decreases food popularity and acceptability among consumers [119]. Use of high throughput molecular technologies and novel treatment techniques for preservation of food products can be useful to reduce the amount of salt in fermented food processing in the food industry. Although alternative techniques, such as gamma-irradiation, microencapsulation, ultrasound, and high-pressure treatment, to prevent spoilage in the fermented foods are in early stages of development, the preliminary results seem promising [131,132,133,134].

## 5. Conclusions

Regular consumption of fermented foods can exert beneficial effects on body weight regulation and metabolic function through several mechanisms. Although the level of evidence for some types of fermented foods is still very low and there are considerable knowledge gaps regarding the molecular mechanisms of action of specific fermented foods, the current findings are promising. This may be especially relevant if personalized dietary interventions can be designed to include individual variation and consider multiple factors like genetic, immune, and neurological variation. Compliance to treatment is a key factor to obtain optimal results in intervention studies, and since most fermented foods are integral in the traditional cuisine in most countries, their use as a drug delivery system may increase chances of success in the long-term. Finally, analyzing omics data in addition to clinical outcomes can result in a deeper understanding of the mechanisms of action and the links among metabolism, immune responses, and eating behavior.

## Figures and Tables

**Figure 1 ijms-24-02665-f001:**
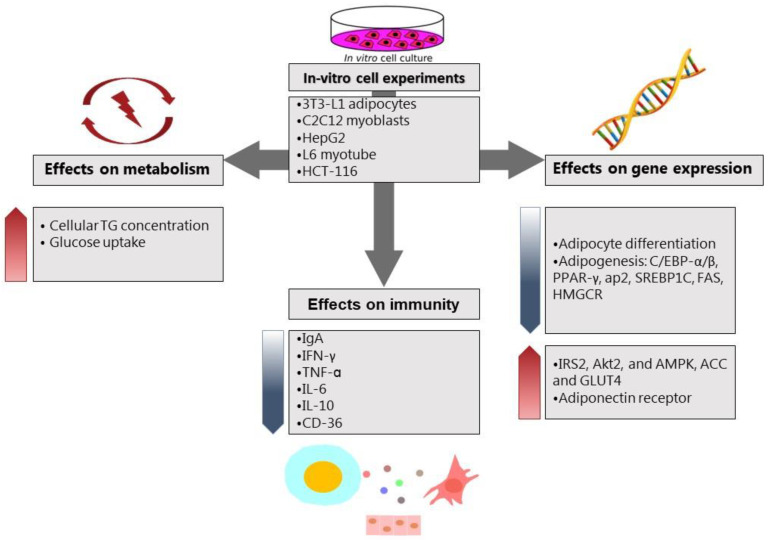
The effects of various fermented foods on metabolism, immunity, and gene expression of metabolic factors in cell culture studies. Abbreviations: TG: triglyceride, IgA: immunoglobulin A, IFN-γ: interferon-γ, TNF-α: tumor necrosis factor-α, IL-6: interleukin-6, IL-10: interleukin-10, CD-36: cluster of differentiation-36, C/EBP-α/β: CCAAT/enhancer-binding protein-α/β, PPAR-γ: Peroxisome proliferator-activated receptor-γ, AP-2: adaptor protein complex-2, SREBP-1c: sterol regulatory element-binding transcription factor 1, FAS: Fas cell surface death receptor, HMGCR: HMG-CoA reductase, IRS-2: insulin receptor substrate 2, AKT-2: serine/threonine kinase, AMPK: AMP-activated protein kinase, ACC: acetyl-CoA carboxylase, GLUT4: insulin-regulated glucose transporter.

**Figure 2 ijms-24-02665-f002:**
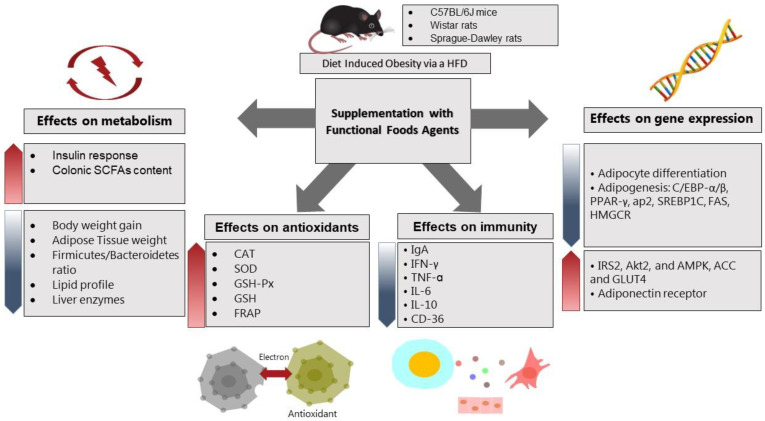
The effects of various fermented foods on metabolism, antioxidant status, immunity, and gene expression of metabolic factors in three different animal models. Abbreviations: SCFAs: short chain fatty acids, CAT: catalase, SOD: superoxide dismutase, GSH-PX: glutathione peroxidase, GSH: glutathione, FRAP: fluorescence recovery after photobleaching, IgA: immunoglobulin A, IFN-γ: interferon-γ, TNF-α: tumor necrosis factor-α, IL-6: interleukin-6, IL-10: interleukin-10, CD-36: cluster of differentiation-36, C/EBP-α/β: CCAAT/enhancer-binding protein-α/β, PPAR-γ: peroxisome proliferator-activated receptor-γ, AP-2: adaptor protein complex-2, SREBP-1c: sterol regulatory element-binding transcription factor 1, FAS: Fas cell Surface death receptor, HMGCR: HMG-CoA reductase, IRS-2: insulin receptor substrate 2, AKT-2: serine/threonine kinase, AMPK: AMP-activated protein kinase, ACC: acetyl-CoA carboxylase, GLUT4: insulin-regulated glucose transporter.

**Table 1 ijms-24-02665-t001:** Summary of the effects of kefir products on obesity and mechanisms of action.

Food/Compound	Treatment, Administration and Duration	Model/Animal	Molecular Mechanisms Relevant to Obesity and Metabolism		
			Metabolic Marker	Inflammation and Oxidative Stress Marker	Ref.
Citrus pomace bioconversion (CPB) supplementation	5 weeks administration of CPB by kefir lactic acid bacteria on a high-fat diet (HFD)	C57BL/6J mice	↓ BW gain, AT weight/BW, TG, and adipocyte diameter ↑ gene expression of UCP-1 and PGC-1α in AT		Youn et al. [38]
Fermented Okara	3 weeks administration of HFD	ICR mice	↓BW gain, AT weight, TG level, and ↓ lipid accumulation in the liver SREBP1 and FAS ↑PPARɑ and ɣ		Ichikawa et al. [39]
Yogurt intake and branched chain hydroxy acids	12-week administration of HFD	C57Bl/6 mice a model of obesity-linked type 2 diabetes	preserved glucose homeostasis and insulin sensitivity, prevented hepatic steatosis preserved claudin-3 gene expression		Daniel et al. [40]
Tempeh	Fermented soybean was administered as breakfast	crossover study on 13 females with 1 week washout	↑Acyl-ghrelin ↑Arginine levels, Insulin response		Noer et al. [41]
Surface layer proteins (SLP) isolated from kefir probiotic lactic acid bacteria	6 weeks administration of 120 mg of DH5 SLP or LCM8 SLP/kg body weight on HFD	C57BL/6J mice	↓ BW gain and AT weight, TG, and insulin resistance 41 genes were downregulated 38 genes were upregulated	IL-6 and production of NF-kB p65 protein	Kim et al. [42]
Fermented mixed grain (FMG) with digestive enzymes	12 weeks administration of high and low dose FMG on HFD	C57BL/6J mice	↑BW, WAT, and plasma lipids, FBS ↑pgc1a and SIRT1 Improved hepatic steatosis	↓ Leptin, keratinocyte chemoattractant, IL-6, TNF-ɑ, and MCP-1↑Adiponectin	Han et al.[43]
Fermented blueberry juice (FBG)	17-week administration of FBG on HFD (Equivalent to 250 mL day^−1^ of orange juice for human)	C57BL/6 mice	↓ BW gain, fat accumulation, serum lipids, insulin resistance, ↑Firmicutes and obesity-related bacteria population Lean bacteria population and caecal SCFA ↑GLUT-4, GCK, LDL-receptor, and PPARα expression ↓ SCD, SREBP1c, and FAS	↓ Leptin, TNF-ɑ ↑ IL-10 and SOD	Zhong et al. [44]
Mixed lactobacilli	8-week administration of 0.2 mL of the mixed lactobacilli suspension (10^8^ CFU) on HFD	C57BL/6J mice	↓TG, TC, and LDL-C, ALT, and AST levels, lipid accumulation ↑ HDL-c, cecal acetic acid, and butyric acid ↑ AMPK and HSL expression ↓ ACC, FAS, PPAR-γ, and C/EBP-ɑ	↓ CAT, SOD, GSH-Px, GSH and MDA	Li et al. [45]
Lactobacillus fermentum (LF) CQPC05, isolated from Sichuan Pickle	8-week administration of 1.0 × 10^9^ CFU/kg of LF-CQPC05 on HFD	SPF C57/BL6J mice	↓ Enlargement of body tissues, fat cell hypertrophy ↓ ALT, AST, TC, TG, and LDL-C and expression level of LPL, PPAR-ɑ ↑ CYP7A1, and CPT1, PPAR-ɣ, and C/EBP-ɑ		Zhu et al.[46]
Lactobacillus acidophilus (La-SJLH001) isolated from fermented food	20-week administration of 100 μL and 10^9^ CFU La-SJLH001 on HFD	C57BL/6J mice	Improved Total cholesterol level and OGTT ↓Pancreatic island size and fat droplets in liver ↑Ileum villi length 308 up-regulated and 536 down-regulated genes		Sun et al.[47]
Ginseng vinegar (GV)	10-week administration of low, medium, and high dose of GV by oral feeding needle on HFD	C57/BL6 mice preventive model and therapeutic model	↓Epididymal fat weights, TG, TC, and LDL-c levels,	↓IL-6 in preventive model and CRP in therapeutic model TNF-ɑ expression in both	Oh et al. [4]
Koji glycosylceramide (KGC) in traditional fermented foods	3-week administration of semisynthetic AIN-76 diet supplemented with 1% KGC	C57BL/6J mice and db/db mice (C57BLKS/J Iar- +Lepr db/+Lepr db)	↑ Bile acids in cecum and feces, ↓ Gene expression of CYP7A1 and ABCG8 ↓ABCG5, FXR, LDL-r, HMGCoA reductase, and SREBP2, serum glucose, liver cholesterol		Hamajima et al. [48]
Fermented green tea extract (FGT)	9-week administration of 400 mg/kg FTG on HFD	3T3-L1 mouse preadipocytes and C2C12 mouse myoblasts C57BL/6 mice	↓Adipocyte differentiation, cellular TG concentrations, PPARɣ, ap2, ↑CC, SREBP1C, FAS expression, ↓BW gain and fat mass CPT1, mCAD, UCP2, tfam ↓Firmicutes/Bacteroidetes ratio and Ruminococcus population	↓CD-36	Cho et al. [49]
Lactobacillus plantarum (Ln4) isolated from cabbage kimchi	4-week administration of 5 × 10^8^ CFU live Ln4 by oral gavage in 0.2 mL of distilled water on HFD	3T3-L1 adipocytes C57BL/6 mice	Inhibits lipid accumulation and weight gainenhances glucose uptake ↑ IRS2, Akt2, and AMPK	↓ANGPT-L3, CRP, Leptin, Lipocalin-2, MCP-1, IGFBPCD-36	Lee et al. [50]
Laminaria japonica Extract (LJF) a seaweed	Cell culture treatment by 100 and 200 μg/mL LJF for 48 h	3T3-L1 adipocyte	↓ C/EBP-α/β, PPAR-γ Adiponectin, glucose uptake and TG production		Kim et al. [51]
Fermented Pumpkin by Bacillus subtilis and Lactobacillus plantarum (Cucurbita moschata extract (FCME))	8-week administration 0.1%, 0.3%, 0.5% FCME on HFD	C57BL/6 mice	↓ BW gain, serum lipids (FFA, TC, TG, LDL), ALT, AST, insulin, and GIP ↓ Expression of PPAR*γ*, C/EBP*α*, *β*, *γ*, and SREBP-1C ↓ Size of adipocytes		Hossain et al. [52]
Foodborne lactic acid bacteria from a fermented dairy product “Mozzarella di Bufala Campana” (MBC)	Orally supplemented for 15 days with 1 × 10^9^ CFU/day of a mixture of natural LAB strains extracted from MBC. Then went on a HFD for 90 days	C57BL/6J mice	↓ Epididymal WAT, TC, TG ↓ Firmicutes/Bacteroidetes ratio	↑ Regulatory T cells and CD4+ ↓ CD8+ cell numbers, IL-6, TNF-ɑ, and IFN-γ	Roselli et al. [53]
Fermented moleifera extract (FM) by Lactobacillus sakei, Lactobacillus plantarum, and Lactobacillus brevis	8-week administration of FM (250 mg/kg body weight in 150 lL DW) on HFD	C57BL/6J mice	↓ Liver weight and fat improved glucose tolerance	↓ Endoplasmic reticulum stress, oxidative stress, and lipotoxicity in quadriceps muscles TNFɑ, IL-6, IL-1ꞵ, and IL-12	Joung et al.[54]
Synthetic acetic acid vinegar and Nipa vinegar	10-week administration of vinegar (0.08 and 2 mL/kg BW) on HFD	C57BL/6 mice	↓ BW and fat pad/BW ratio Improved TC, TG, LDL, HDL and leptin ↑ GLUT2 and SREBP1 expression adipose GLUT4 adiponectin expression ↓ Firmicutes/Bacteroidetes ratio ↑ Verrucomicrobia and Proteobacteriaphylum	↑ SOD, GSH, and FRAP ↓ NF-κB and iNOS	Beh et al. [55]
Yogurt With or Without Probiotic	8-week administration of probiotic yogurt provided by Bifidobacterium animalis (10^8^ CFU/mL)	Mice	↓ BW, glucose, total, LDL and HDL-cholesterol and TG Improved Thymus histology ↑ Bifidobacteria and Lactobacilli population	↑ IgA+, IFNγ, IL-6, IL-10	Balcells et al. [56]
Fermented Rhizoma Atractylodis Macrocephalae (FRAM)	2-week administration of 250 mg/kg FRAM contains 7 × 10^9^ CFU/mL Lactobacillus plantarum on HFD (100–400 mg/mL for invitro assays)	HepG2, L6, and 3T3- L1 cell lines Male Sprague- Dawley rats	↓ adipose tissue weight, TG, AST levels, expression of C/EBP-a and HMGCR, NO production, HFD+LPS-induced endotoxemia ↑ Insulin sensitivity, expression of AMPK, gut Bifidobacterium and Akkermansia	↓ CRP, TNF-ɑ, and IL-6	Wang et al. [57]
Fermented soybean extract by Bacillus subtilis (BTD-1)	1 week treatment with 10, 50 and 100 μg/mL BTD-1	3T3-L1 preadipocytes	↓Lipid accumulations and C/EBPα expression ↑ ACC protein and GLUT4, ↑ glucose uptake into the adipocytes ↓ PPARγ (not sig)		Hwang et al. [58]
Fermented soybean paste (cheonggukjang) (CKB)	8-week administration of 30% CKB provided by Bacillus licheniformis-67 on HFD	C57BL/6 mice	↓ BW, epididymal fat pad weight, TC, FBS, leptin and Insulin, liver X receptor ɑ↑ CPT-1		Choi et al. [59]
Rice koji produced by Aspergillus oryzae and Aspergillus kawachii	4-week administration of HFD containing 10% (w/w) of rice koji powder (white, yellow, and red)	C57BL/6J mice (L6 myotube cells) muscle cells	↓ Weight gain, epididymal white adipose tissue, and total adipose tissue Weight, Leptin ↑ Adiponectin ® and GLUT4 expression		Yoshizaki et al. [60]
Fermented Flos Lonicera (FFL) by Lactobacillus plantarum	8-week administration of 250 mg/kg of FFL containing 2 × 10^7^ CFU/mL Lactobacillus plantarum on HFD+LPS	RAW 264.7 and HCT-116 cell lines Sprague-Dawley rats	↓ Body and adipose tissue weights, TC, HDL, TG, AST, and endotoxin levels in serum, urinary lactulose/mannitol ratio, and lipid accumulation in liver modify Akkermansia and ratio of Bacteroidetes and Firmicutes	↓ NO production TNF-ɑ, COX-2, and IL-6	Wang et al. [57]
Fermentable carbohydrate (FC)	8-week administration of FC inulin on HFD	C57BL/6 mice	↑ Weight gain, food intake, and AMPK activity Colonic acetate		Frost et al. [61]
Fermented Kochujang (FK) (soybean product)	12-week administration of HFD containing 22% FK	C57BL/6 mice	↓ Weight gain, epididymal fat, TC, TG, and glucose level, ACC expression ↑ ACS, CPT-1, and UPC-1		Koo et al. [62]
Fermented Lowbush blueberry juice by Serratia vaccinii bacterium	6-h treatment of cells with 30 mmol/L GAD Fermented compound	C2C12 myotubes 3T3-L1 adipocytes	↑ glucose uptake PPARɣ and AMPK activity		Vuong et al. [63]
Whole milk and fermented milk	2-day (washout 3 months) administration of 1.4 L milk or fermented milk, plus 165 mg Acetate for a 75 kg man	Randomized crossover design Eight healthy men	Slower gastric emptying greater increase and a quicker decrease of the triacylglycerol content in all lipoprotein fractions and of CCK, GIP, GLP-1, and PYY		Sanggaard et al. [64]
Kefir culture broth	(0.1 mg mL^−1^) for 2–4 days	3T3-L1 mouse preadipocyte cell lines	↓ C/EBPα, PPARγ, and SREBP-1c aP2, FAS, and ACC and GPDH activity	TNF-α	Ho et al. [65]
Kefir	3 weeks oral administration of 0.2 mL of kefir milk twice a day 9.62 ± 0.19 Log CFU of lactic acid bacteria, 9.52 ± 0.12 Log CFU of acetic acid bacteria, and 7.67 ± 0.30 Log CFU of yeast per mL	Female BALB/c mice	↓ Firmicutes, Proteobacteria, and Enterobacteriaceae ↑ Lactobacillus, Bacteroidetes acetic acid bacteria, and total yeast in fecal samples		Kim et al. [66]
Kefir strains, Lactobacillus mali APS1 (APS1) and L. kefiranofaciens M1 (M1)	1. 8 weeks intra-gastrically Feeding HFD + 10^8^ CFU/mouse L. kefiranofaciens M1 Or 10^8^ CFU/mouse Lactobacillus mali APS1 (APS1) 2. Caco-2 monolayers were co-cultured with 10^6^–10^8^ CFU/mL of L mali APS1	Diet-induced obese mice	↑ GLP-1 and insulin sensitivity ↓ BUN, TC, LDL-C, TG, serum insulin Improved Clostridia/Bacteroidia ratio TEER of polarized Caco-2 monolayers chemokine CCL-20	↓ Th1 (TNF-α and IL-6) ↑IL-10 cytokines	Lin et al. [67]
Kefir powder, lactic acid bacterial number greater than 10^8^ CFU/g, a yeast cell number greater than10^2^ CFU/g, and 50 mg/kg polysaccharides	8 weeks on a 0.1% or 0.2% kefir powder-supplemented HFD	Male C57BL/6 J mice	↑ Adipocyte size in epididymal and liver fat PPARγ, C/EBPα and β, aP2, SREBP-1c, FAS, and ACC TC, LDL-C, ALT, AST	↓TNF-α and IL-6	Choi et al. [68]
Kefir drink	8 weeks, two servings/d kefir drink	75 women aged 25 to 45 years	↓ TC, LDL-C, TG non-HDLC, TC/HDLC LDLC/HDLC		Fathi et al. [69]
Isolated lactic acid bacteria (LAB) from kefir	2 × 10^8^ CFU of Lactobacillus kefir DH5 supplemented HFD	Male C57BL/6 mice	↓ PPAR-γ, FABP4, and CPT1 expression, TC, LDL-C, TG ↑HDL-C Changes in gut microbiota		Kim et al. [70]
Kefir milk contained 9.84 ± 0.36 log CFU/mL of lactic acid bacteria and 7.23 ± 0.41 log CFU/mL of yeast	12-week orally administered 0.2 mL of kefir milk	Male C57BL/6 mice	↑PPARα and AOX ↓ACC, SREBP-c1, DGAT ACS TC, LDL-C, TG, Leptin Modified HDL-C gut microbiota	↓IL-6, MCP1, SOD2	Kim et al. [71]
Exopolysaccharide isolated from kefir grains	4 weeks HF diets containing 5%BG, 5% EPS, or 8% kefir-grain residue obtained after EPS	Male C57BL/6 mice	↓TC Modified HDL-C, LDL-C, VLDL-C gut microbiota		Lim et al. [72]
Traditional kefir	12-week oral gavage of 100 uL of kefir in HFD	Wild type C57BL/6 female mice	Modified TC and liver TG ↓FGF-15 expression of FAS, PPARγ ↑Cyp7a1 gut microbiota		Bourrie et al. [73]
Isolated kefir peptides	8-week orally administered kefir peptides 164 mg/kg in HFD	Male Sprague-Dawley rats	↓LDL-C, FFA, and FAS expression ↑*p*-ACC ↑pAMPK, PPAR-α, and CPT-1 expression	↓TNF-α, IL-6, IL-1β, and TGF-β	Tung et al. [74]
Tibet kefir milk	8-week oral gavage of 18 mL/kg BW TKM	Female Sprague-Dawley rats	↑BW, TG LPL and ATGL in fat and LPL in liver and serum gut microbiota		Gao et al. [75]
Isolated kefir lactic acid bacteria + Whole Grape Seed Flour	9 weeks supplementation of WGF + LAB and LAB in HF-diet-fed mice	Diet-induced obese (DIO) mice	Improved fecal microbiota profiles		Seo et al.[76]
Kefir soymilk fermentation	90 days HFD and additional 10 mL/kg of body weight of a KSF by gavage	Wistar female rats	↓ Lipase and α-amylase activities Improved lipid profile and glucose		Tiss et al. [77]
Traditional and pitched kefir with the full complement of microbes or not	HFD supplemented with 2 ml in 20 g of food for 8 weeks	Female C57BL/6 mice	Improved lipid profile liver TG levels, PPARγ	IL-1ꞵ IL-8 (Nonsignificant)	Bourrie et al. [73]
Tibet kefir milk	8 weeks of gavage volume of 18 mL/kg BW in HFD	Female Sprague-Dawley rats	Modified metabolic pattern of amino acids and gut microbiota metabolites in favor of anti-obesity effects		Gao et al.[78]
Kefir drink	Contains 10^7^ CFU/g probiotic bacteria (Lactobacillus spp. and Streptococcus spp.)	healthy females aged 21–24 years	↓Postprandial appetite		Caferoglu et al.[79]
Paraprobiotic Kefir Lactic Acid Bacteria (PLAB)	6-week oral administration of 120 mg/kg body weight PLAB in HFD	C57BL/6J mice Macrophage RAW 264.7 cells	Improved lipid profile and glucose ↓Insulin and HOMA-IR and expression of 41 genes ↑expression of 38 genes	IL-6 NF-κB	Kim et al. [42]
Surface layer protein (SLP) exopolysaccharides (EPS) grape seed flour (GSF)	6-week orally administered 250 mg/kg EPS, or 120 mg/kg SLP or saline with fed 2% GSF (GSF) or combination (42 mg/kg EPS + 20 mg/kg SLP + 0.5% GSF; ALL)	C57BL/6J mice	Glucose tolerance test (GTT) Insulin tolerance test (ITT) Improved lipid profile Fecal microbiota change ↓Expression of 91 genes	Fgf13 Kng1 Oxtr, Ptafr, Serpina3m	Seo et al. [80]

Abbreviations in Table 1 in alphabetic order: ABCG = ATP Binding Cassette Subfamily G, ACC = acetyl CoA carboxylase, ACS = acyl-CoA synthetase, ACSL-1 = Acyl-CoA synthetase long-chain family member 1, Akt2 = protein kinase Bβ, ALT = Alanine transaminase, AMPK-α = Adenosine 50-monophosphate-activated protein kinase-α, ANGPT-L3 = Angiopoietin-like 3, AOX = Alternative oxidase, AST = Aspartate transaminase, AT = adipose tissue, ATGL = Adipose triglyceride lipase, BW = body weight, CAT = Catalase, CCK = cholecystokinin, CD = cluster of differentiation, C/EBP = CCAAT-enhancer-binding proteins, CFU = Colony Forming Unit, COX-2 = Cyclo-oxygenase 2, CPT-1 = carnitine palmitoyl transferase-1, CRP = C-reactive protein, CYP7A1 = Cytochrome P450 Family 7 Subfamily A Member 1, DGAT = Diglyceride acyltransferase, FABP4 = F atty acid binding protein4, FAS = Fatty acid synthetase, FBS = Fasting blood sugar, FFA = Free fatty acid, FGF = Fibroblast Growth Factor, FXR = Farnesoid X receptor, GCK = Glucokinase, GLUT = glucose transporter, GLP-1 = glucagon-like polypeptide-1, GIP = gastric inhibitory polypeptide, GPDH = Glycerol-3-phosphate dehydrogenase, GSH-Px = Glutathione peroxidase, GTT = Glucose tolerance test, HDL-C = High Density Lipoprotein Cholesterol, HFD = High Fat Diet, HSL = Hormone sensitive lipase, IGFBPs = insulin-like growth factor binding proteins, Ig = Immunoglobin, IL = Interleukin, INF-γ = Interferon-γ, iNOS = Intrinsic nitric oxide synthesis, IRS = insulin receptor substrate, ITT = insulin tolerance test, LAB = lactic acid bacteria, LDL-C = Low Density Lipoprotein Cholesterol, LDL-R = LDL receptor, Lep-r = Leptin receptor, LPL = lipoprotein lipase, LPS = Lipopolysaccharide, mCAD = Medium-chain acyl-CoA dehydrogenase, MCP-1 = Monocyte chemoattractant protein-1, MDA= Malondialdehyde, NO= Nitric oxide, OGTT= oral glucose tolerance test, Oxtr = Oxytocin receptor, PGC1 α = Peroxisome proliferator-activated receptor gamma coactivator 1-alpha, PPAR = peroxisome proliferator-activated receptor, Ptafr = Platelet Activating Factor Receptor, PYY = peptide YY, SCD = Stearoyl-CoA desaturase, SCFA = short-chain fatty acids, Sirt 1 = Sirtuin 1, SLP = surface layer proteins, SOD = superoxide dismutase, SREP1C = Sterol regulatory element-binding transcription factor 1, TEER = transepithelial resistance, TG = Triglyceride, TGF-β = Transforming growth factor beta, Th1 = T helper1, TNF-α = tumor necrosis factor-α, UCP-1 = uncoupling protein-1, VLDL-C = Very Low-Density Lipoprotein Cholesterol, WAT = White adipose tissue.

## Data Availability

Not applicable.

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
