# Peer review of "Fermented Foods in the Management of Obesity: Mechanisms of Action and Future Challenges"

_ijms, 2023, doi:10.3390/ijms24032665_

Round 1
Reviewer 1 Report
The presented manuscript is focused on the health properties of fermented food. Especially its influence on obesity was discussed. Although the topic is interesting I have some comments and remarks.
Firstly, the article is wrongly prepared and not in the accordance with the guideline for authors.
There are no keywords in this paper
The chapters are not numbered.
References are inadequately formatted.
What for the authors included double times the lists of abbreviations? If necessary, it should be included in the beginning without such a widely spread table.
Reference should be cited in rectangular brackets
Ect….
Moreover, the included template should be used to prepare the manuscript.
Please include the aim of the work in the abstract.
The numbering of subchapters is wrong.
Figures 1 and 2 have no caption. These figures should be also described not only mentioned.
In conclusion, the authors have repeated commonly known information. Please modify them and give what is new according to this review.
Table 1 should be inserted in the body of the manuscript. Line 136. Not after the conclusion section.
Author Response
Dear reviewer, thank you for your useful comments. Please see our answers to the comments below each comment.
- The presented manuscript is focused on the health properties of fermented food. Especially its influence on obesity was discussed. Although the topic is interesting, I have some comments and remarks.
Reply: Thank you for your kind comment; we appreciate your time and effort that has helped us improve the quality of our manuscript.
2- Firstly, the article is wrongly prepared and not in the accordance with the guideline for authors.
Reply: We apologize for this. We have thoroughly revised our manuscript according to the journal’s guidelines.
3- There are no keywords in this paper
Reply: We added the keywords at the end of the abstract.
4- The chapters are not numbered.
Reply: We added numbering to the chapters of our manuscript.
5- References are inadequately formatted.
Reply: We formatted the list of references according to the journal style.
6- What for the authors included double times the lists of abbreviations? If necessary, it should be included in the beginning without such a widely spread table.
Reply: This was because we listed all abbreviations used in the manuscript at the beginning of the paper, and also listed the abbreviations used in the table as a footnote to the table (so the table could be independently read). We have now only listed the abbreviations in the table footnote. All other abbreviations in the text are defined on first mention, per journal’s instructions.
7- Reference should be cited in rectangular brackets
Reply: We changed the format of the references according to the journal style.
Ect….
8- Moreover, the included template should be used to prepare the manuscript.
Reply: We used the provided Word template for our revised manuscript. However, in order for you and the Editor to be able to see the exact changes we made, we also submitted a “tracked changes” version of the original submission as supplementary material.
9- Please include the aim of the work in the abstract.
Reply: We now include the aim our paper in the abstract.
10- The numbering of subchapters is wrong.
Reply: We changed all the numbers of the chapters and subchapters according to the journal template.
11- Figures 1 and 2 have no caption. These figures should be also described not only mentioned.
Reply: We added captions for figures 1 and 2 and now we mentioned them in the text.
12- In conclusion, the authors have repeated commonly known information. Please modify them and give what is new according to this review.
Reply: We were invited to contribute a review paper on the health effects of fermented foods; by definition, therefore, our paper does not provide original new data. We have amended the “conclusion” section to highlight future research perspectives, but also possible risks of consumption of fermented foods (requested by reviewer 2).
13- Table 1 should be inserted in the body of the manuscript. Line 136. Not after the conclusion section.
Reply: We inserted Table 1 and Figures 1 and 2 in the body of the manuscript.
Reviewer 2 Report
Dear Editor, in my opinion, this review is hard to read being really not fluid. Furthermore, the journal format is not followed in the in-text citations and table. Just one big table was reported, this fact also complicates the ms structure.
Dealing with conceptual contents, the ms highlighted only the benefits of fermented food and what about the risks?
The authors must deeply review the ms before publication.
Author Response
Dear reviewer, thank you for your useful comments. Please see our answers to the comments below each comment.
- Dear Editor, in my opinion, this review is hard to read being really not fluid. Furthermore, the journal format is not followed in the in-text citations and table. Just one big table was reported, this fact also complicates the ms structure.
Reply: Thank you for all useful comments. We appreciate your time and effort that helped us improve the quality of the manuscript.
- Dealing with conceptual contents, the ms highlighted only the benefits of fermented food and what about the risks?
Reply: In the revised manuscript we describe the challenges and risks associated with the consumption of fermented foods (in sections 2-4). In addition, we added a relevant sentence in the abstract.
- The authors must deeply review the ms before publication.
Reply: We thoroughly revised the manuscript to abide by the formatting requirement of the journal. We hope these changes improved the quality of the manuscript to a satisfactory level.
Round 2
Reviewer 1 Report
The authors corrected the manuscript as suggested. However, in the future, all changes should be marked in the manuscript in a different colour or correct in the review mode.
Author Response
Dear reviewer,
Thank you for your kind comments. We will consider your comments to highlight the changes for future manuscripts.
Reviewer 2 Report
Revise citations in Table 1
Author Response
Dear Reviewer,
Thank you for your time and comment. We revised the citation in Table 1.